# An Insulin Upstream Open Reading Frame (INSU) Is Present in Skeletal Muscle Satellite Cells: Changes with Age

**DOI:** 10.3390/cells13221903

**Published:** 2024-11-18

**Authors:** Qing-Rong Liu, Min Zhu, Faatin Salekin, Brianah M. McCoy, Vernon Kennedy, Jane Tian, Caio H. Mazucanti, Chee W. Chia, Josephine M. Egan

**Affiliations:** Intramural Research Program, National Institute on Aging, National Institutes of Health, 251 Bayview Blvd, Baltimore, MD 21224, USA; zhumi@grc.nia.nih.gov (M.Z.); bmccoy6@asu.edu (B.M.M.); tianj2@mail.nih.gov (J.T.); caio.mazucanti@nih.gov (C.H.M.); chiac@grc.nia.nih.gov (C.W.C.); eganj@grc.nia.nih.gov (J.M.E.)

**Keywords:** insulin, INSR, skeletal muscle, satellite cells, isoforms

## Abstract

Insulin resistance, stem cell dysfunction, and muscle fiber dystrophy are all age-related events in skeletal muscle (SKM). However, age-related changes in insulin isoforms and insulin receptors in myogenic progenitor satellite cells have not been studied. Since SKM is an extra-pancreatic tissue that does not express mature insulin, we investigated the levels of insulin receptors (INSRs) and a novel human insulin upstream open reading frame (INSU) at the mRNA, protein, and anatomical levels in Baltimore Longitudinal Study of Aging (BLSA) biopsied SKM samples of 27–89-year-old (yrs) participants. Using RT-qPCR and the MS-based selected reaction monitoring (SRM) assay, we found that the levels of INSR and INSU mRNAs and the proteins were positively correlated with the age of human SKM biopsies. We applied RNAscope fluorescence in situ hybridization (FISH) and immunofluorescence (IF) to SKM cryosections and found that INSR and INSU were co-localized with PAX7-labeled satellite cells, with enhanced expression in SKM sections from an 89 yrs old compared to a 27 yrs old. We hypothesized that the SKM aging process might induce compensatory upregulation of INSR and re-expression of INSU, which might be beneficial in early embryogenesis and have deleterious effects on proliferative and myogenic satellite cells with advanced age.

## 1. Introduction

Skeletal muscle satellite cells (MuSCs) reside beneath the basal lamina of myofibers and are mitotically quiescent under sedentary conditions [1]. However, upon strenuous exercise and muscle injury, quiescent MuSCs proliferate and differentiate to replace damaged myofibers and replenish the MuSCs quiescent pool [2]. The capability of MuSCs to replenish myofibrils declines progressively during muscle aging due to concurrent inflammation, senescence, and apoptosis [3]. Peripheral insulin resistance [4], dysregulation of glucose metabolism [5], impairment of insulin signaling [6], and aberrant expression of alternatively spliced insulin receptor (INSR) isoforms [7] contribute to SKM age-related features.

Human-specific genes and isoforms [8,9] are developmentally regulated and disproportionally more active in human fetal cortex [10], and many are inactivated after birth [11]. Insulin has very important roles in cell survival and apoptosis during gastrulation and neurulation in the developing embryo, and dysregulation of its expression causes embryopathies [12]. Its dysregulation can occur with increasing age. Recently, we uncovered a novel human insulin upstream open reading frame (uORF) that could be transcribed and translated into INSU isoforms in pancreatic islet β-cells and choroid plexus [13]. The INSU uORF translational initiation methionine is 45 bp upstream of the conventional insulin mRNA 5′-cap site, and the 5-prime untranslated region (5′UTR) of INSU overlaps with the conventional promoter region of the insulin primary open reading frame (pORF), which includes the proximal glucose-responsive *cis*-acting elements of the TATA, G, A, C, E, and CRE boxes that bind to the critical insulin transcription factors, PDX1, MAFA, NEROD1, and CREB [14]. We previously found that both insulin and INSU were expressed in epithelium cells of the choroid plexus, in which insulin secretion is regulated by serotonin and not glucose metabolism [13,15]. We hypothesized that human INSU with intrinsic disorder regions (IDRs) participates in embryonic cellular differentiation and is suppressed postpartum, but then reemerges with age in extra-pancreatic tissues. We developed a rabbit polyclonal antibody for INSU and confirmed the presence of INSU in pancreatic β-cells in human islets. We found that INSU mRNA and protein were also expressed in non-insulin-expressing human skeletal muscle samples, and its expression increased with age in MuSCs.

## 2. Materials and Methods

### 2.1. Clinical Biopsy of Skeletal Muscles

Skeletal muscle biopsies (medial gastrocnemius, age 27–89 yrs, *n* = 48) were obtained from participants of an IRB-approved clinical study, The Baltimore Longitudinal Study of Aging, performed by IRP/NIA at MedSTAR Harbor Hospital (NIH IRB # 03-AG-N322).

### 2.2. RNA Isolation, cDNA Synthesis, and RT-qPCR

Total RNAs were extracted from the human skeletal muscle (SKM) using TRIzol reagent (Cat#15596026, Thermo Fisher Scientific, Waltham, MA, USA) protocol. Single-strand cDNA was synthesized from total RNA using qScript™ XLT cDNA SuperMix (Cat# 95161-100, QuantaBio, Beverly, MA, USA). INSU-specific TaqMan probes were designed specifically for the exon-1 and intron-1 retention junction (probe: CCATCAAGCAGGTCTG-Fam; forward: GGGAGATGGGCTCTGAGACTATAA; reverse: AGCCCACCTGACGCAAAG). We designed probes for human INSR-A (probe: TCTCGGAAACGCAGGTC-Fam; forward: CTGCACAACGTGGTTTTCGT; reverse: GATGTTGGGAATGTGACGGTG) and INSR-B (probe: ACCCTAGGCCATCTC-Fam; forward: CTGCACAACGTGGTTTTCGT; reverse: AAACGCAGGTCCCTTGGC) isoforms and a pan-insulin probe specifically to the exon-2 that is included in all insulin pORF isoforms (probe: TGAACCAACACCTGTGCG-Fam; forward: CCCAGCCGCAGCCTTT; reverse: AGAGAGCTTCCACCAGGTGTGA) for the TaqMan assay. TaqMan probes of PAX7 (Hs00242962_m1), NLRP3 (Hs00918082_m1), MyHC-emb (MYH3, Hs01074230_m1), insulin-like growth factor 2 (IGF2, Hs01005964_m1), and endogenous control glyceraldehyde-3-phosphate dehydrogenase (GAPDH, Vic-labeled, Cat#4326317E) were ordered from Thermo Fisher Scientific (Waltham, MA, USA). The duplex fluorescent TaqMan assay was performed in replicates, and the relative fold change was calculated using the formula: 2^−△△Ct^ formula for all the probes. For more accurate absolute quantification of low abundant INSU mRNA in SKM samples, INSU was normalized with endogenous control β2 microglobulin (B2M, Vic-labeled, Cat# 4326319E) in droplet digital PCR (ddPCR) to avoid abundant endogenous probes, such as GAPDH saturation of the positive droplets in Poisson distribution of positive and negative droplets (QX200 ddPCR System, Bio-Rad, Philadelphia, PA, USA).

### 2.3. Sample Preparation and MS Based-Selected Reaction Monitoring (SRM)

The SKM biopsies were rinsed three times with ice-cold 1x PBS containing 1x HaltTM Protease Inhibitor Cocktail (Cat# 78438, Thermo Fisher Scientific, Halethorpe, MD, USA) to remove blood, and then homogenized in 300 μL of 0.1% RapiGest (Cat# 186001861, Waters Corp. Milford, MA, USA) containing 0.1 M Tris-HCl (pH = 8.0) on ice twice for 10 s each, pausing for 30 s between each homogenization, followed by 30 min on ice. After centrifugation at 16,800× *g* for 20 min at 40 °C, 150 μg of total protein was loaded into an automated liquid handling platform for sample dilution, denaturation, reduction, alkylation, and trypsin digestion, followed by semi-automated solid-phase extraction of tryptic digests, as described in detail elsewhere [16]. The signature peptide selection was based on an empirical procedure that balances the ideal attributes of the assays with practical limitations, for instance, the tryptic peptides available from the INSU. The selected peptides (Table 1) were synthesized as synthetic stable isotope (heavy)-labeled and unlabeled analogs by Genemed Synthesis Inc. (San Antonio, TX, USA). After reconstituting peptides, the concentration of each synthetic peptide was determined by amino acid analysis (New England Peptide, Gardner, MA, USA). SRM parameters require optimization and refinement from tuning acquisition to select the best fragmentation parameters and the best transitions for the final SRM assay. We selected the best charge state, declustering potential (DP), collision energy (CE), and collision cell exits potential (CXP) for each peptide. The final signature peptides were selected by analyzing the SRM chromatogram of an isotopically labeled version of the peptide and comparing it to the mixed peptide pool from proteolytically digested pooled SKM samples. To further enhance SRM sensitivity, we scheduled the mass spectrometer to collect subsets of peaks based on the target analyte retention times (RT) on the column. Three to six interference-free precursor and fragment masses for a given peptide (transition) constituted the final SRM assay of INSR (pep-INSR) and PAX7 (pep-PAX7) (Table 1 and Figure 1A,B). The insulin B-chain (pep-B1), INSU (pep-U2 and pep-U4) were derived from our previous work [13]. We used another INSU SRM pep-U2 to confirm the pep-U4 results. Compared with classical SRM modes, the scheduled SRM provides better signal-to-noise due to higher dwell times and greatly improved reproducibility and accuracy by detecting more data points across chromatographic peaks. Details of SRM quantitative validation are shown in our previous publication [17].

### 2.4. RNAscope Fluorescent In Situ Hybridization (FISH)

The RNAscope FISH probe for INSU was custom-designed (11 ZZ pairs targeting 2-732 of nucleotide sequence of MT335690 of INSU in C2 channel). The FISH probes of INSR (Cat# 450061) and PAX7 (Cat# 546691) in C1-channel were ordered from cataloged probes from Advanced Cell Diagnostics Inc. (ACD, Hayward, CA, USA). Cryosections of frozen skeletal muscle biopsies, pretreatment, probe hybridizations, amplifications, and labeling were performed according to RNAscope Multiplex Fluorescent Detection Kit v2 protocol, as previously described [13]. Images were acquired using a Zeiss LSM 880 confocal microscope (Carl Zeiss AG, Baden-Württemberg, Germany).

### 2.5. INSU Antibody Development, Immunohistochemistry, and Amyloid Staining

The INSU epitope (QKRPSSRSVPRAFAS) was conjugated to keyhole limpet hemocyanin (KLH) as an antigen, and the anti-INSU rabbit polyclonal antibody was custom-made and affinity-purified by Genemed Synthesis Inc. (San Antonio, TX, USA). Frozen biopsied skeletal muscle sections were fixed using 4% PFA for 15 min and washed 3 times in PBS for 5 min each. Hydrophobic barriers were then drawn using the ImmuoEdge Pen (Cat# H-4000, Vector Lab, Bulingame, CA, USA). The blocking buffer was prepared using 5% Normal Goat Serum, 0.3% Triton X-100, and PBS. After the PBS wash, the sections were blocked for 1 h. After blocking, the skeletal muscle sections were incubated with rabbit polyclonal antibody INSU and mouse monoclonal antibody PAX7 (MAB1675, Bio-Techne, Minneapolis, MN, USA). After the PBS washes, the secondary antibodies of Alexa-Flour 647 anti-rabbit and Alexa-Flour 568 anti-mouse were used for staining to avoid autofluorescence on a 488 nm wavelength. After the PBS washes, the same secondary antibodies were used to stain the sections as above. Images were taken using a Zeiss 880 confocal microscope, as described previously [15]. Congo-Red staining was performed following the manufacturer’s protocol (Cat# PS108, FD NeuroTechnologies, INC. Columbia, MD, USA). Images were acquired using a Axiovert Observer microscope (Carl Zeiss AG, Baden-Württemberg, Germany).

### 2.6. Statistical Data Analysis

GraphPad Prism v9.0.1 software was used for statistical analysis, and data are presented as means ± SEM. The normalized expression values of INSU isoforms TaqMan fold changes and ddPCR droplets, as well as the SRM relative ratio of endogenous peptide targets (L, light) and stable isotope-labeled peptides (H, heavy), were analyzed with the two-tailed unpaired *t*-test, one-way ANOVA, and two-tailed paramedic Pearson’s correlation. *p* < 0.05 was considered significant.

## 3. Results

### 3.1. INSU Promoter cis-Elements

The INS gene promoter region (~400 bp) upstream of the INS transcription start site (TSS) contains *cis*-elements that bind to transcription factors that control insulin β-cell specific expression and glucose response [14]. These *cis*-elements are more conserved in primate than other vertebrates [14]. We found that the TSS of INSU is localized at −272 relative to the TSS of INS (+1); therefore, the 5′UTR of INSU overlaps with some INS *cis*-elements, including the Z, C2, E2, A3, A2, E1, A1, and G1 boxes and cAMP response elements (CRE1 and CRE2) [18] that bind to PDX1 and MAFA and are required for β-cell maturation, survival, and glucose sensing [19]. However, the promoter region of INSU retains an insulin kelobase upstream INK box, variable tandem repeats (VNTR), the SP1 site, the A5 box, and an enhancer core that induce INSU expression in β-cells (Figure 2A). The INSU translational initiation site contains a consensus human Kozak ribosomal binding site with a score of 0.87 [20] (Figure 2B) that allows for the translation of INSU in β-cells [21].

### 3.2. Positive Correlation of INSR, INSU, and PAX7 in Skeletal Muscle Biopsies with Age

We investigated INSR, INSU, and PAX7 age-related mRNA and protein levels in SKM samples from participants aged 27–89 yrs (*n* = 48). At the mRNA level, INSU (*n* = 13) was 33.3% of INSR-A (*n* = 15), and insulin (using the pan insulin probe) was below the detection in our SKM biopsy samples (Figure 3A). We found that INSU mRNA (r = 0.323, *p* = 0.025) (Figure 3B) and protein levels (r = 0.357, *p* = 0.009) (Figure 4A) were positively correlated with age, and the pep-U4 peptide correlation was confirmed by pep-U2 (r = 0.29, *p* = 0.04). INSR-A and -B isoform (Figure 3C,D) mRNA levels (r = 0.662, *p* < 0.0001; r = 0.665, *p* < 0.0001, respectively) and INSR protein (r = 0.356, *p* < 0.015) (Figure 4B) levels were also positively correlated with age, as were the satellite cell marker PAX7 mRNA (r = 0.374, *p* = 0.009) (Figure 3E) and protein (r = 0.316, *p* = 0.020) (Figure 4C) levels, the inflammasome NLRP3 mRNA level (r = 0.367, *p* < 0.014) (Figure 3F), myosin heavy chain-embryonic MyHC-emb mRNA (r = 0.291, *p* = 0.027) (Figure 3G), and IGF2 (insulin-like growth factor 2) mRNA (r = 0.394, *p* = 0.006) (Figure 3H). Interestingly, we identified a very low level of insulin B-chain peptide below the limit of quantification (LOD) that was significantly increased in SKM with age (Figure 4D). Since we did not find insulin mRNA in SKM, the detected insulin B-chain peptide might be insulin bound to INSR in SKM.

### 3.3. Localization of INSR and INSU mRNA in Myogenic Satellite Cells and Inflammasome

Since the RT-qPCR and SRM proteomics showed upregulation of INSU at both the mRNA and protein levels, we used representative SKM sections from a young and an old participant to identify cellular localization of INSU. We carried out Congo red amyloid staining on SKM sections from participants aged 27 and 89 yrs. In the 27 yrs section, we observed minimal to no Congo red positivity (Figure 5A), while in the 89 yrs section, we detected numerous Congophilic stains that were potential amyloid deposits on the periphery of myofibrils (Figure 5B). We then performed duplex RNAscope FISH on the frozen sections from the 27 yrs section and observed the colocalization of INSR in PAX7-labeled satellite cells (white arrow, Figure 5C), as well as in other myofibers (green arrows). Moreover, interestingly, we found an increase in INSR mRNA in PAX7-labeled satellite cells (white arrows, Figure 5D) in the SKM section from the 89 yrs section. We observed a small amount of INSU in PAX7-labeled satellite cells in the SKM section from the 27 yrs participants (Figure 5E) but found abundant INSU and PAX7 increased in the SKM section from the 89 yrs participant (white arrows, Figure 5F). Furthermore, we investigated the colocalization of INSU with inflammasomes and found that only a few INSU- and NLRP3-labeled cells were present in the SKM section from the 27 yrs participant (white arrow, Figure 5G), but prominently increased in NLRP3-labeled myofibers in the SKM section from the 89 yrs participant (white arrows, Figure 5H).

### 3.4. Identification of INSU Protein in SKM Myogenic Satellite Cells

We previously found that insulin was secreted from choroid plexus epithelium cells independent of glucose, but dependent on serotonin stimulation [15]. We then extended our INSU immunofluorescence study to non-insulin-producing SKM biopsies and found that the INSU protein was colocalized with PAX7-labeled satellite cells in the SKM section (Figure 6A). We observed that the numbers of the colocalized cells increased in the SKM section of the 89 yrs participant (Figure 6B, white arrows), agreeing with INSU SRM proteomic results.

## 4. Discussion

We found an insulin upstream open reading frame isoform INSU to be present in myogenic satellite cells, and its expression was progressively increased in SKM biopsies with age. INSU isoforms are hominid-specific [13], similar to hominid-specific IAPP (Islet Amyloid Polypeptide) isoforms, which evolved to protect β-cells and adapt to a human-specific metabolism [22]. The CpG sites in the INS promoter region are fully methylated in embryonic stem cells and gradually demethylated when they become β-cells but remain largely methylated in non-β-cells [23]. The INSU 5′UTR overlaps with the INS promoter; therefore, the INSU promoter has less CpG methylation regulation, and its expression has less cell-type restriction. The SKM INSU expression might be due its truncated promoter, which lacks a major transcription factor and the epigenetic regulations that are progressively compromised with aging. SKM is responsible for postprandial insulin-mediated glucose disposal, and insulin signaling is a major contributor to maintaining the homeostasis of myofibers, both qualitatively and quantitatively [7]. During myogenesis, MuSCs undergo proliferation in the early stages and differentiation in the later stages, which is regulated by IRS1/AKT/mTOR pathway [2]. Heparan sulfate proteoglycans, e.g., syndican-3 (SDC3), complexes with INSR in response to insulin, and the up-regulation and down-regulation of SDC3 promotes MuSC proliferation and differentiation, respectively [24]. An increase in INSR in aged SKM MuSCs may be a cellular attempt to compensate for defective homeostasis and insulin resistance in myofibrils in old age. The INSU isoform might regulate insulin secretion from β-cells and change the insulin and INSR interaction that plays an important role in MuSCs proliferation during SKM aging.

Sporadic inclusion-body (s-IBM), lymphocyte infiltration, T cell inflammation, and inflammasomes are often associated with myofiber degeneration in aging [25,26]. Misfolded proteins in s-IBM resemble β-pleated-sheet amyloid proteins in neurodegenerative diseases, such as amyloid-β, pTau, and α-synuclein [27]. INSU might be misfolded because it contains intrinsic disorder regions (IDRs) and potential disorder-to-order transition. The machine learning algorithm FuzPred [28] predicted that INSU would contain regions of multivalency with disorder-to-order probabilities (PDO) of 0.572, and FuzDrop [29] predicted three aggregation hot spots at the N-terminal region of INSU. Elevated INSU in MuSCs during aging might increase the amyloid burden and serve as a neoantigen to induce inflammation [30].

The extra-pancreatic INSU expression might play dual hormonal and growth roles in the embryo and again in aging. In embryos, insulin has roles in cell survival and apoptosis during differentiation, and unprocessed proinsulin itself is required for embryonal survival [31]. Human and Aves species diverged about 600 MYA [32], yet chicken embryonic-specific insulin uORF controls low-level and glucose-independent expression of proinsulin not only in the neuroepithelial cells of the ectoderm, but also in the mesoderm and endoderm layers [33]. Similar to chicken embryo-specific insulin mRNA, which has an uORF active in early embryogenesis [33], INSU isoforms likely play a role in early embryo development [9] independent of glucose uptake that is suppressed in extra-pancreatic tissues after birth but reemerges in aged tissues. Since we observed the co-upregulation and co-localization of INSU with myo-satellite cells in SKM samples with age, we hypothesized that the embryonic INSU isoforms’ over-expression in aged stem cells might inhibit stem cell proliferation and regeneration.

Verdijk et al. reported that satellite cell content decreases together with type II fiber size, but not type I fiber size, with the main data points clustered in the age groups of 20–25 and 65–80 yrs [34], and our data points were collected in SKM biopsies of participants aged 27–89 yrs. PAX7 plays a role in early-embryo development of gastrulation and neurulation through interactions with other developmental genes [35]. Overexpression of PAX7 pushes MuSCs toward a quiescence state and prevents the activation of MuSCs [36,37]; INSU may interact with PAX7, thereby further inhibiting the activation of MuSCs in aging SKM. The promoter sequence of INSU is a truncated version of the conventional INS gene promoter, and INSU might have wider cell-type presence than insulin, which is most abundant in pancreatic β-cells. INSU might play different roles in different cell types. We previously found that INSU protein levels were reduced in islets of T2D participants [21], and we now demonstrate that the INSU protein is positively correlated with SKM age. Pancreatic islets and SKM are derived from the embryonic endoderm [38] and mesoderm [39] layers, respectively, where insulin plays an important trophic role in the development of embryo layers [31]. In the current study, we found that the genes involved in embryogenesis and development, INSU, INSR, PAX7, MyHC-emb, and IGF2 [40], as well as the inflammasome gene NLRP3, increased with SKM age, indicating that epigenetic restrictions of cell-type specific expression after birth [23] are compromised in SKM with age.

## 5. Conclusions

A novel human insulin upstream open reading frame is present in aging skeletal muscles, and its localization in myogenic progenitor satellite cells negatively impact SKM regenerative capacity.

## Figures and Tables

**Figure 1 cells-13-01903-f001:**
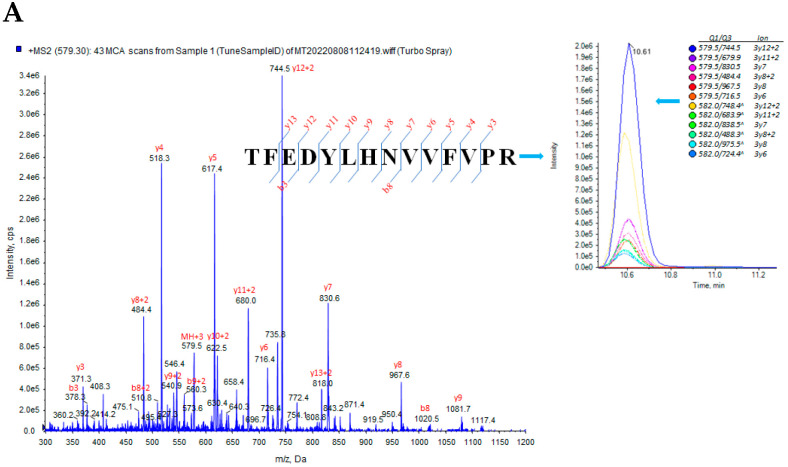
SRM peptide transition peaks. (**A**) Insulin receptor pep-INSR, (**B**) PAX7 isoform-3 pep-PAX7 in SKM matrix. The peptide cleavage sites are marked by Z signs on the left panel, and Q1/Q2 ion ratio in SKM matrix is on the right panel.

**Figure 2 cells-13-01903-f002:**
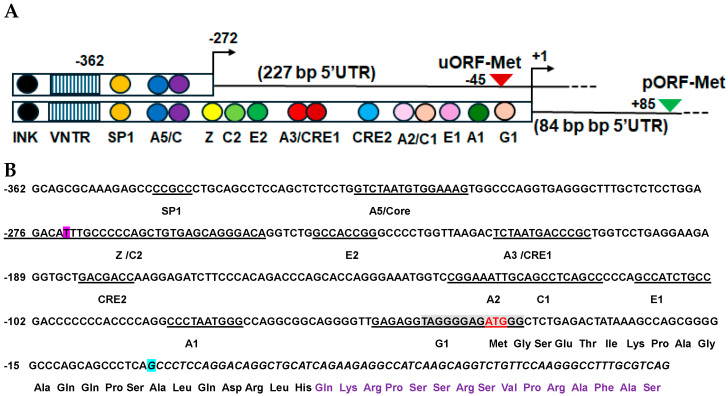
INSU and INS promoter *cis*-elements: (**A**) open rectangular boxes represent promoters; solid and dashed lines, mRNA transcripts; red and green triangles, INSU and INS translational initiation sites with the relative nucleotide positions of uORF-Met (red triangle) and pORF-Met (green triangle), respectively; arrows, transcription start sites; INK, insulin kilobase upstream box, striped boxes, variable number tandem repeats (VNTR); SP1, transcription factor SP1; A5/C, overlapping A5 box and enhancer core; Z, glucose sensing Z box; C2 box; E2 box; A3/CRE1, overlapping A3 box and cAMP response element site 1; CRE2, cAMP response element site 2; A2/C1, neighboring A2 and C1 boxes; E1 box; A1 box; and G1 box. The sizes of 5′UTRs of INSU and INS are in parentheses. INS transcription start site is labeled +1 bp; INSU transcription start site, −272 bp; VNTR, −362 bp. (**B**) INSU and INS promoter and 5′UTR nucleotide sequences downstream of VNTR and upstream of exon 1B splicing site. The negative numbers on the left are nucleotide positions relative to INS transcription start site. The *cis*-element sequences are underlined and labeled with *cis*-element names. INSU TSS is highlighted with pink, Cozak consensus sequence is highlighted with gray, translational initiation codon (ATG) is highlighted with red lettering, INS TSS is highlighted with light blue, and INS 5′UTR sequence is highlighted with italic letters. INSU-coded amino acids are marked under the uORF, and the epitope amino acids are marked by purple lettering.

**Figure 3 cells-13-01903-f003:**
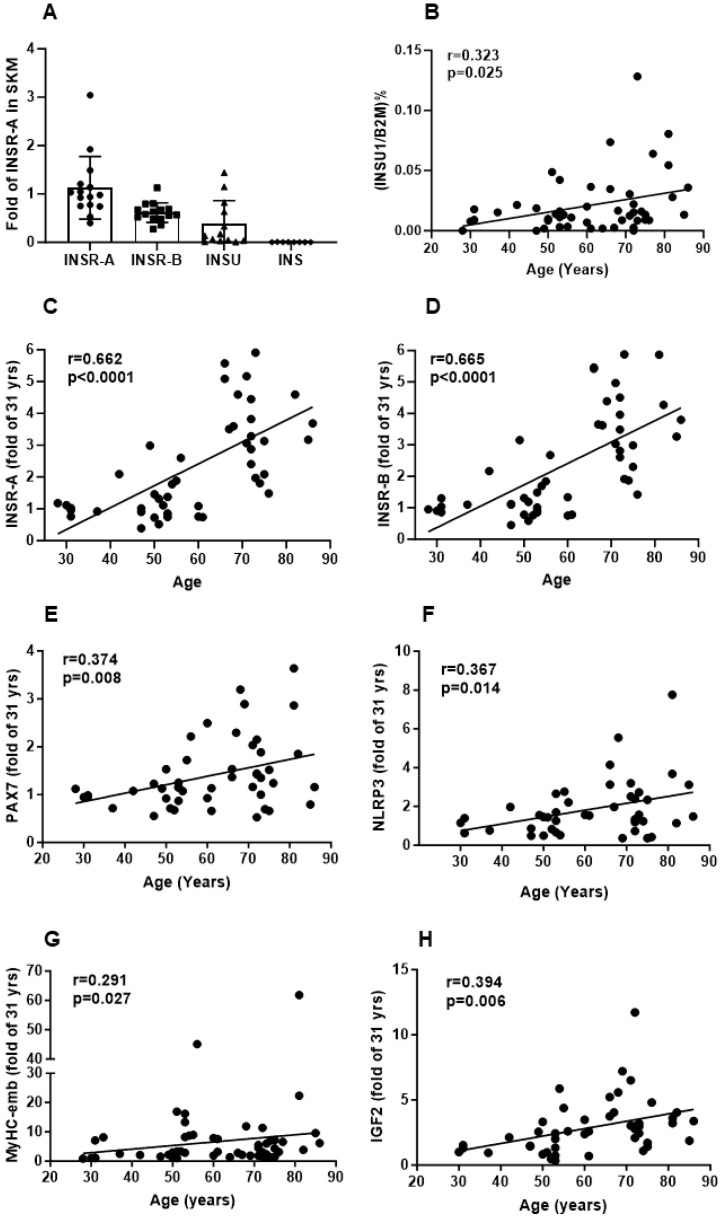
mRNA expression in skeletal muscle samples. (**A**) Comparison of INSR-A, INSR-B, INSU, and insulin mRNA levels using INSR-A as reference in SKM samples. (**B**) Correlation of the absolute quantification of INSU mRNA levels with age. Correlation of the relative quantification of INSR-A (**C**), INSR-B (**D**), PAX7 (**E**), NLRP3 (**F**), MyHC-emb (**G**), and IGF2 (**H**) mRNA levels with age, using 31 yrs as a reference.

**Figure 4 cells-13-01903-f004:**
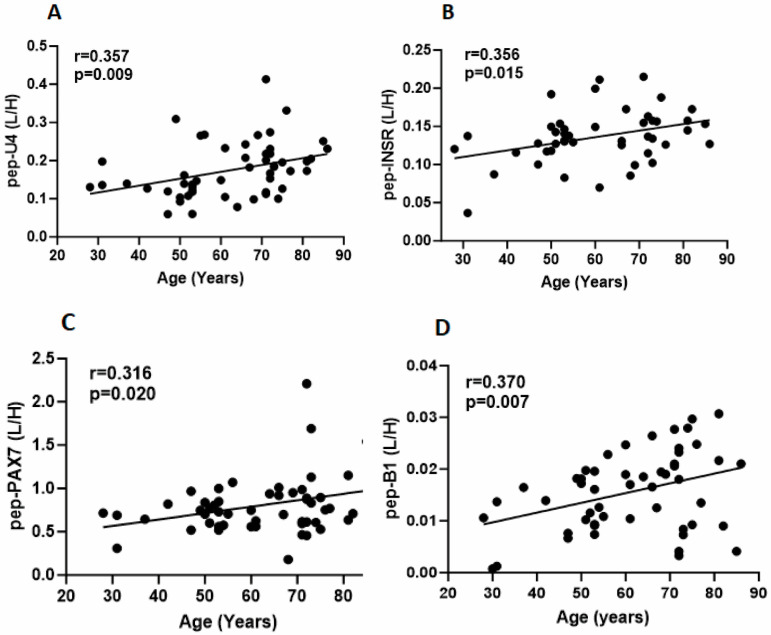
Protein expression of INSU (**A**), INSR (**B**), PAX7 (**C**), and insulin B-chain (**D**) in skeletal muscle samples. Y-axis is the ratio of the unlabeled peptide (L) and the stable isotope labeled peptide (H), and X-axis is the age in years.

**Figure 5 cells-13-01903-f005:**
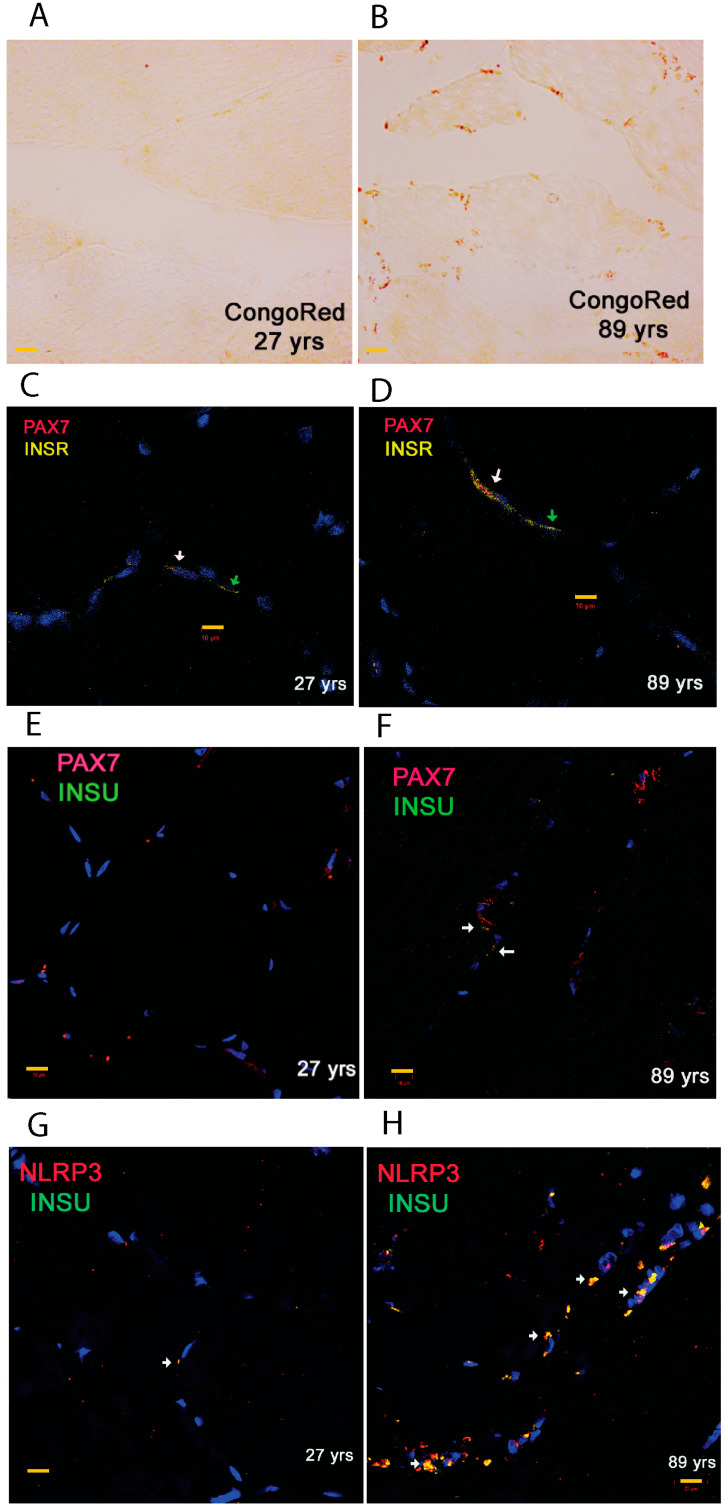
Congo red stain and RNAscope fluorescence in situ hybridization (FISH) of SKM sections. Congo red stains of SKM sections of participants aged 27 yrs (**A**) and 89 yrs (**B**) in which red stains mark the potential amyloid deposits. PAX7/INSR duplex FISH of participants aged 27 yrs (**C**) and 89 yrs (**D**), PAX7/INSU duplex FISH of participants aged 27 yrs (**E**) and 89 yrs (**F**), NLRP3/INSU duplex FISH of participants aged 27 yrs (**G**) and 89 yrs (**H**) in the muscle. White arrows represent the overlapping signals, and green arrows indicate non-overlapping signals. Scale bars are marked with yellow in 10 μm.

**Figure 6 cells-13-01903-f006:**
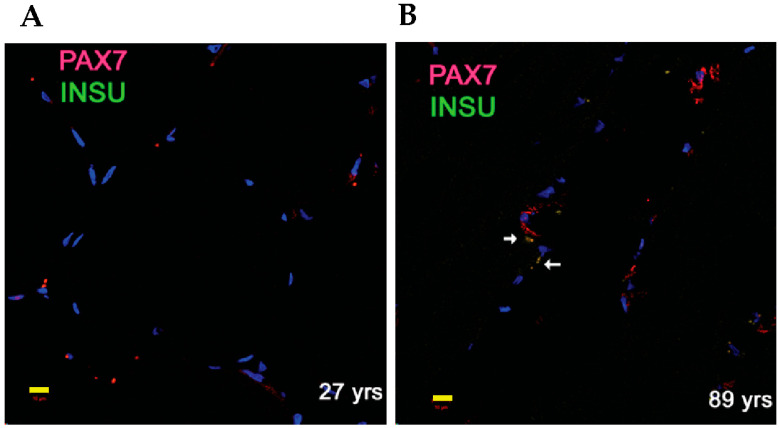
Duplex immunofluorescence staining of INSU and PAX7 in SKM sections from samples of 27 yrs (**A**) and 89 yrs (**B**). White arrows represent overlapping signals of INSU and PAX7 and yellow bar, 10 μm.

**Table 1 cells-13-01903-t001:** Molecular weights (MWs) of unlabeled (light) and stable isotope-labeled (heavy) trypsin peptides used for SRM assay. The labeled amino acids are marked by red lettering and ^ signs. [CAM] represents carbamidomethylation of cysteine residues to prevent the disulfide bond formation and other oxidative changes.

Peptide Name	Peptide Sequences	MW (dal)	AA
pep-INSR_light	H2N-TFEDYLHNVVFVPR-OH	1735.9	14
pep-INSR_heavy	H2N-TFEDYLHNVVFV^PR-OH	1743.9	14
pep-PAX7_light	H2N-NVSLSTQR-OH	904.5	8
pep-PAX7_heavy	H2N-NVSLSTQR^-OH	914.5	8

## Data Availability

INSU rabbit polyclonal antibody and the unlabeled and stable isotope-labeled peptides for the LC-MS/MS SRM assay are available upon reasonable request from the corresponding authors.

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
