# Peer review of "An Insulin Upstream Open Reading Frame (INSU) Is Present in Skeletal Muscle Satellite Cells: Changes with Age"

_cells, 2024, doi:10.3390/cells13221903_

Round 1

Reviewer 1 Report

Comments and Suggestions for Authors

The authors report on the biochemical characterization of a recently described (Liu et al, 2021) stretch in the hnMRNA (INSU) of insulin leading to an mRNA species which is designated 'upstream open reading frame isoform' (which represents an mRNA isoform with alternatively spliced exon upstream of the so far known exon-1 if I understood the paper correctly). The study is focused on the INSU expression in skeletal muscle biopsies from non-diseased study participants between the ages of 27 and 89. For quantification of protein levels in tissue homogenates, mass spectrometry-based selected reaction monitoring (SRM) with synthetic stable/unstable isotopes was applied. For localization of INSU in skeletal muscle sections, fluorescent in situ hybridization (FISH) was performed. Bioinformatic comparison of gene sequences from databases showed INSU is expressed in humans and chimpanzees, but not in other monkey species. In the skeletal muscle homogenates of older study participants, significantly higher and correlated levels of INSU and PAX7 (satellite cell markers) were expressed than in those of younger study participants. FISH demonstrated stronger histological expression of INSU signals in satellite cells of a biopsy from an 89-year-old than of a biopsy from a 27-year-old participant.

The project is interesting. The manuscript is well written but of reduced quality hampered by some significant drawbacks in the presentation.

Major objections:

1. Too many superfluous information is represented which which makes the reading confusing for the reader, e.g. Figure 1 (washing effect of biopsies) is trivial and should be deleted or Fig. 3 B-D showing the gene sequence comparison for insulin B-chain, insulin-receptor and PAX7 in several primates which are not required for understanding the project.

2. It is commonly accepted that the number of satellite cells decreases in correspondence to the age or the degree of sarcopenia (e.g. PMID: 24122288, PMID: 31122394)However, in this article, the authors report higher concentrations of PAX7 over age without discussing this clear contradiction to the literature.

Minor objections:

1. An abbreviation list would be helpful.

2. Line 35/36: What is the significance of the statement that neuronal tissue is obviously a 'playground' of evolution? The cells/tissues investigated in this paper (skeletal muscle, satellite cells) are not of neuronal origin.

3. Line 39/40: I am confused about the terminology 'insulin upstream open reading frame (uORF) isoform, INSU. Isn't this DNA stretch simply a new, untranslated exon?

4. Lines 136ff: Why is the immunohistochemical methodology described if this method is not applied to the sections? (c.f. lines 213ff)

5. Lines 182-184 belong to the discussion.

6. Lines 213: It is stated that histology was performed on biopsy sections from participants in the age 27-89 but shown are examples only from a 27 y and a 89 y subjects. Without quantitative data, the impression is created that only the two samples were analyzed. Avoid suggesting that more samples were analyzed or indicate how many samples were examined histologically. This statement is essential for assessing the quality of the discussion.

Author Response

Open Review 1

Comments and Suggestions for Authors

The authors report on the biochemical characterization of a recently described (Liu et al, 2021) stretch in the hnMRNA (INSU) of insulin leading to an mRNA species which is designated 'upstream open reading frame isoform' (which represents an mRNA isoform with alternatively spliced exon upstream of the so far known exon-1 if I understood the paper correctly). The study is focused on the INSU expression in skeletal muscle biopsies from non-diseased study participants between the ages of 27 and 89. For quantification of protein levels in tissue homogenates, mass spectrometry-based selected reaction monitoring (SRM) with synthetic stable/unstable isotopes was applied. For localization of INSU in skeletal muscle sections, fluorescent in situ hybridization (FISH) was performed. Bioinformatic comparison of gene sequences from databases showed INSU is expressed in humans and chimpanzees, but not in other monkey species. In the skeletal muscle homogenates of older study participants, significantly higher and correlated levels of INSU and PAX7 (satellite cell markers) were expressed than in those of younger study participants. FISH demonstrated stronger histological expression of INSU signals in satellite cells of a biopsy from an 89-year-old than of a biopsy from a 27-year-old participant.

The project is interesting. The manuscript is well written but of reduced quality hampered by some significant drawbacks in the presentation.

Major objections:

  1. Too many superfluous information is represented which which makes the reading confusing for the reader, e.g. Figure 1 (washing effect of biopsies) is trivial and should be deleted or Fig. 3 B-D showing the gene sequence comparison for insulin B-chain, insulin-receptor and PAX7 in several primates which are not required for understanding the project.

We deleted Figure 1 and figure 3 B-D accordingly.

  1. It is commonly accepted that the number of satellite cells decreases in correspondence to the age or the degree of sarcopenia (e.g. PMID: 24122288, PMID: 31122394).  However, in this article, the authors report higher concentrations of PAX7 over age without discussing this clear contradiction to the literature.

We discussed this controversy at the end of the Discussion, before Conclusions, and cited the references that you suggested.

Thank you for your insight.

Minor objections:

  1. An abbreviation list would be helpful.

We added an abbreviation list. Thank you for your suggestion.

  1. Line 35/36: What is the significance of the statement that neuronal tissue is obviously a 'playground' of evolution? The cells/tissues investigated in this paper (skeletal muscle, satellite cells) are not of neuronal origin.

In the introduction line 35/36 “Human-specific genes and isoforms are developmentally regulated and disproportionally more active in human fetal cortex and many are inactivated after birth.” We added “its dysregulation could occur in old age” to emphasize the extra-pancreatic INSU in SKM aging.

  1. Line 39/40: I am confused about the terminology 'insulin upstream open reading frame (uORF) isoform, INSU. Isn't this DNA stretch simply a new, untranslated exon?

The upstream open reading frame is the extended exon of the insulin exon-1 (PMID: 34649926) that can be transcribed and translated into INSU peptides. To reduce the confusion, we added in the introduction “Recently we uncovered a novel insulin upstream open reading frame (uORF) that could be transcribed and translated into INSU isoforms in pancreatic islet b-cells and choroid plexus”. We replaced the Fig 1 with a new diagram of INSU promoter showing cis-elements, transcription start site, 5’UTR, and translational initiation site in comparison with those of INS gene.

  1. Lines 136ff: Why is the immunohistochemical methodology described if this method is not applied to the sections? (c.f. lines 213ff)

We have the immunohistochemistry INSU IF figures of SKM sections in the revised Fig 6.

  1. Lines 182-184 belong to the discussion.

We deleted line 182-184 to clarify the results.

  1. Lines 213: It is stated that histology was performed on biopsy sections from participants in the age 27-89 but shown are examples only from a 27 y and a 89 y subjects. Without quantitative data, the impression is created that only the two samples were analyzed. Avoid suggesting that more samples were analyzed or indicate how many samples were examined histologically. This statement is essential for assessing the quality of the discussion.

Thank you for the suggestion. We used only one 27 yrs and one 89 yrs SKM sections as representative IF to show cell type specificity of INSU since we have already quantitative mRNA and peptide levels of INSU by RT-qPCR and SRM. We added “Since the RT-qPCR and SRM proteomics showed the upregulation of INSU at both mRNA and protein levels, we used a representative young and an old SKM sections to identify cellular localization of INSU.”

Submission Date

30 August 2024

Date of this review

06 Sep 2024 06:52:55

Open Review 2

Comments and Suggestions for Authors

  1. At least 2 peptides should be used to quantify the epitope level in SRM assay.

In Method we added “We used another INSU SRM pep-U2 to confirm pep-U4 results.”. In Results we added “and confirmed by another INSU SRM pep-U2 (r=0.29, p=0.04)”.

  1. Beside showing the imaging results of FISH and IF experiments, statistic analysis should be presented as well.

We answer the similar question of the reviewer 1. We used only one 27 yrs and one 89 yrs SKM sections as representative IF to show cell type specificity of INSU since we have already quantitative mRNA and peptide levels of INSU by RT-qPCR and SRM. We added “Since we have the upregulation of INSU at both mRNA and protein levels, we used a representative young and an old SKM sections to identify cellular localization of INSU.”

  1. The author should explain the relationship between the level of INSR (or INSU) and aging process.

We added at the end of first paragraph of the Discussion “INSU isoform might regulate insulin secretion from b-cells and change insulin and INSR interaction that plays an important role in MuSCs proliferation during SKM aging.”

Comments on the Quality of English Language

  1. “p” in p value should be shown as italic.

We changed the p to italic p. Thank you for the correction.

  1. Redundant blank space should be removed.

Yes, we removed the redundant blank spaces. Thank you for the correction.

Submission Date

30 August 2024

Date of this review

12 Sep 2024 06:15:3

Editor’s comment September 24, 2024

It came to our attention that the main text is short. We kindly invite you to enrich the content of your manuscript.

We added a new correlation of embryonic SKM fiber markers MyHC-emb and IGF2 with SKM age in Fig 3G and H to demonstrate the transition of aged SKM fibers to embryonic fibers.

The new text is labeled with red font.

Reviewer 2 Report

Comments and Suggestions for Authors

1. At least 2 peptides should be used to quantify the epitope level in SRM assay.

2. Beside showing the imaging results of FISH and IF experimets, statistic analysis should be presented as well.

3. The author should explain the relationship between the level of INSR (or INSU) and aging process.

Comments on the Quality of English Language

1. “p” in p value should be shown as italic.

2. Redudant blank space should be removed.

Author Response

(The authors gave the same response as above.)

Round 2

Reviewer 1 Report

Comments and Suggestions for Authors

The authors have improved the study satisfactorily